# Derivation and validation of the BIMAST score for predicting the presence of fibrosis due to Metabolic dysfunction-associated steatotic liver disease among diabetic patients in the community

**Roberta Forlano**[1], **Tijana Stanic**[2], **Sahan Jayawardana**[2], **Benjamin H. Mullish**[1], **Michael Yee**[3], **Elias Mossialos**[2,4], **Robert D. Goldin**[5], **Salvatore Petta**[6], **Emmanouil Tsochatzis**[7], **Mark R. Thursz**[1], **Pinelopi Manousou**[1]*

1 Liver Unit/Division of Digestive Diseases, Department of Metabolism, Digestion and Reproduction, Faculty of Medicine, Imperial College London, London, United Kingdom, 2 Department of Health Policy, London School of Economics and Political Science, London, United Kingdom, 3 Section of Endocrinology and Metabolic Medicine, St Mary's Hospital, Imperial College NHS Trust, London, United Kingdom, 4 Centre for Health Policy, The Institute of Global Health Innovation, Imperial College London, London, United Kingdom, 5 Department of Cellular Pathology, Faculty of Medicine, Imperial College London, London, United Kingdom, 6 Section of Gastroenterology and Hepatology, PROMISE, University of Palermo, Palermo, Italy, 7 Institute for Liver and Digestive Health, University College London, London, United Kingdom

* p.manousou@imperial.ac.uk

## Abstract

### Background & aims

Current screening pathways, developed from tertiary care cohorts, underestimate the presence of Metabolic-dysfunction associated steatotic liver disease (MASLD) in patients with type 2 diabetes mellitus (T2DM) in the community. We developed, validated, and assessed cost-effectiveness of a new score for screening the presence of fibrosis due to MASLD in primary care.

### Methods

Consecutive T2DM patients underwent screening for liver diseases with transient elastography (TE). Based on predictors of significant/advanced fibrosis, we generated the BIMAST score (based on aspartate aminotransferase (AST) and body mass index (BMI)) and validated it internally and externally (Royal Free Hospital, London and Palermo Hospital). For cost-effectiveness analysis, 6 screening strategies were compared against standard of care: BIMAST score, ultrasound plus abnormal liver function tests, FIB-4, NAFLD fibrosis score, ELF and transient elastography (TE). A Markov model was built based on fibrosis status. Cost per quality-adjusted life year (QALY) gained and the incremental cost-effectiveness ratio (ICER) were estimated over a lifetime.

**Data Availability Statement:** All relevant data are within the paper and its Supporting Information files.

**Funding:** RF is the recipient of the European Association for the Study of the liver (EASL) PhD fellowship Juan Rodes 2018. BHM was the recipient of an NIHR Academic Clinical Lectureship (award number: CL-2019-21-002) and is now the recipient of a Medical Research Council (MRC) Clinician Scientist Fellowship (MR/Z504002/1)The Division of Digestive Diseases receives financial support from the National Institute of Health Research (NIHR) Imperial Biomedical Research Centre (BRC). The funders had no role in study design, data collection and analysis, decision to publish, or preparation of the manuscript.

**Competing interests:** The authors have declared that no competing interests exist.

## Results

Among 300 patients enrolled, 64% (186) had MASLD and 10% (28) other causes of liver disease. In the whole population, patients with significant fibrosis, advanced fibrosis, and cirrhosis due to MASLD were 17% (50/287), 11% (31/287), and 3% (8/287), respectively. In primary care, BIMAST performed better than other non-invasive markers at predicting significant and advanced fibrosis. Moreover, BIMAST reduced false negatives from 54% (ELF) and 38% (FIB-4) to 10%. In both validation cohorts, BIMAST performance was as good as FIB-4. In the cost-utility analysis, ICER was £2,337.92/QALY for BIMAST.

## Conclusion

The BIMAST predicts the presence of significant fibrosis in the community, reduces false negatives and is cost-effective. The BIMAST score should be included in the holistic assessment of diabetic patients.

## 1. Introduction

Metabolic-dysfunction associated steatotic liver disease (MASLD) represents the leading cause of chronic liver disease globally and the commonest cause of abnormal liver function tests (LFTs) worldwide [1]. Overall, MASLD includes a wide spectrum of liver disorders, ranging from simple steatosis to hepatocellular injury and/or inflammation (Metabolic-dysfunction associated steato-hepatitis, MASH) and a variable degree of fibrosis through to cirrhosis [2]. It now established that fibrosis stage represents the strongest predictor of clinical outcomes in these patients [3, 4]. In patients with MASLD, type-2 diabetes mellitus (T2DM) represents an independent predictor of advanced fibrosis/cirrhosis and of more progressive disease, especially in younger ages [5, 6].

Primary care clinicians play an essential role in identifying patients with MASLD who are at risk of advanced liver disease and may require further evaluation in a specialist setting. Specifically, diagnosing and managing MASLD is still perceived as a challenging task by the general practitioners (GPs) in the UK [7] as well as worldwide [8], mainly due to the lack of clarity around screening and referral/managing pathways. Moreover, diagnosing MASLD in primary care currently relies on the presence and entity of abnormal ALT and sometimes AST values, leaving a considerable proportion of those with significant/ advanced liver disease with normal LFTs undiagnosed in the community [9].

The European Association for the Study of the Liver (EASL) and the American Association for the study of the liver (AASLD) guidelines recommend screening for MASLD in high risk-groups (i.e. patients with metabolic syndrome) following a 2-tier system. Specifically, patients should be stratified using non-invasive markers of fibrosis such as FIB-4 and/or NAFLD fibrosis score in primary care, followed by Enhanced liver fibrosis (ELF) and/or transient elastography in a specialist setting [10]. Interestingly, a recent study has shown that up to two/thirds of the new referrals to MASLD liver clinics are discharged once they had received their first assessment, confirming that current risk-stratification requires implementation [11, 12]. Furthermore, our group has also recently demonstrated that screening with non-invasive markers of fibrosis with FIB-4 and ELF carry high false negative rates, missing a significant portion of young patients with normal LFTs [13]. Notably, both FIB-4 and ELF were historically derived

by tertiary care cohorts and, therefore, may underestimate the disease burden in a low-prevalence disease setting, such as primary care.

In this study, we aimed to develop and validate a new score for screening liver disease with advanced fibrosis secondary to MASLD in primary care. We also assessed the cost-effectiveness of using the new score versus other screening strategies.

## 2. Materials and methods

### 2.1 Study population

This cross-sectional study prospectively recruited consecutive patients with T2DM in primary care and community clinics from the North-West London GP network. All patients were recruited between the 1st April 2019 and 31st January 2020. Retrospective data were accessed at the end of enrolment on the 1st February 2020. All subjects were screened with ultrasound (US), blood tests and transient elastography (TE). Of note, this study population was previously published [13]. More details on screening procedures are available in **S1 Appendix**. MASLD was defined as presence of steatosis with at least one cardiometabolic risk factor and in absence of other causes of liver disease. Significant liver disease was defined as LSM≥8.1 kPa.

### 2.2 Cost-effectiveness analysis

**2.2.1 Screening strategies and identification rates.** In this study, six screening strategies were compared against standard of care: 1) US plus LFTs, 2) FIB-4, 3) NAFLD fibrosis score, 4) BIMAST, 5) ELF, 6) TE. Definition of standard of care for MASLD screening in primary care was obtained from previously published economic evaluations on MASLD screening [14, 15], and defined based on what is routinely performed in the community (either abnormal LFTs or no screening) (**S1 Table in S1 Appendix**). Abnormal LFTs were assumed to prompt referral to hospital, with a 65% specificity and 35% sensitivity for liver fibrosis [12].

In the first tier of each strategy, patients were divided into two groups: no disease/MASLD without significant fibrosis vs MASLD with significant and advanced fibrosis. No disease and MASLD without significant fibrosis were considered the same group for this analysis as the management would be similar and would not trigger referral to secondary care compared to MASLD with significant and advanced fibrosis that triggers referral to specialist care [10]. In strategy 1 (US plus LFTs), MASLD with significant fibrosis was defined as evidence of steatosis and features of chronic liver disease on ultrasound, plus elevated LFTs. In strategy 2, significant fibrosis was defined as FIB-4 >1.3 and in strategy 3, as NAFLD fibrosis score >-1.45. In strategy 4 (BIMAST) significant fibrosis was defined as BIMAST> 0.063, in strategy 5 as ELF≥9.8 and and in 6 (TE) as LSM ≥8.1 kPa [10].

**2.2.2 Decision-analytic model.** We developed a decision tree to characterise the risk stratification and diagnostic performance of each of the primary care screening strategies evaluated [12–14] to characterise the disease pathways of patients based on their initial risk stratification as previously published [13].

The model was built upon four disease states:

1. Mild or no liver disease (MLD) if MASLD with LSM≤8 kPa or there was no disease;

2. Significant and advanced liver disease (SLD) if LSM≥8.1 kPa;

3. Compensated cirrhosis (CC) (histological or biochemical/ radiological evidence of cirrhosis without evidence of decompensation);

4. False negatives if LSM ≥8.1 kPa but were within normal range at screening with other non-invasive markers of fibrosis.

The model was built upon the hypothesis that patients with advanced stages of liver disease could progress to end-stage liver disease, including decompensated cirrhosis (DC), hepatocellular carcinoma (HCC), liver transplant (LT) and death [14]. With a disease status being diagnosed (status of significant liver disease and/or compensated cirrhosis), we hypothesized that there is a probability that the management of the patient is modified to reduce the risk of progression to CC, decompensation or death. In this case, progression rates of significant/advanced liver disease and CC groups were assumed as slower, compared to those whose diagnosis was missed at screening (false negatives) [14]. Details on model input parameters are given in **S1 and S2 Tables in S1 Appendix**.

**2.2.3 Model outcomes.** The cost-utility analysis in the base-case was conducted over a lifetime horizon and generated the cost per quality-adjusted life year (QALY) gained. A discount rate of 3.5% per year was applied to outcomes and costs, as recommended by NICE guidelines [16]. We calculated the average cost-effectiveness and the incremental cost-effectiveness ratio (ICER) compared to standard of care [12, 14]. Life expectancy, lifetime costs, and number of correct diagnoses were also estimated. According to NICE guidelines, a cost-effectiveness threshold (CET) of £20,000/QALY gained was set for the base-case analysis as per previous studies [17]. Key input parameters with the highest level of uncertainty (i.e. transition probabilities, utility values, costs and screening ratios) were varied to determine the impact of their variability on cost-effectiveness results. Sensitivity analysis ranges and probabilistic distributions were derived from previous literature and are reported in **S3-S8 Tables in S1 Appendix**.

## 2.3 External cohorts

Two centres provided a retrospective cohort of patients with T2DM diagnosed with NAFLD as external validation cohorts for the score derived from this population: the Royal Free Hospital (London, United Kingdom) and the Palermo University Hospital (Palermo, Italy).

In both cohorts, MASLD was diagnosed based on either US or histology, and other causes of liver disease were excluded, including use of steatogenic drugs and chronic alcohol consumption > 14 UI per week [18]. Anthropometric parameters, blood tests and TE measurements were recorded for all the patients.

The Royal Free Hospital cohort, included consecutive patients with T2DM who were firstly referred from primary care to liver specialist service through the Camden and Islington pathway [19]. As such, this group included patients selected from primary care based on FIB-4>1.3 and/or ELF>9.5 (selected primary care population). The Palermo Hospital cohort included consecutive patients with T2DM who were followed-up in the specialist NAFLD clinic (tertiary care population).

## 2.4 Ethical approval

All patients' recruitment was conducted in line with Good Clinical Practice and sample handling according to Human Tissue Act regulations. The main cross-sectional study (derivation cohort) obtained full ethical approval from the Research Ethics Committee (REC approval 18/LO/1742, IRAS 251274) and all patients included in the study provided written consent form. Data collected retrospectively were fully anonymised; all patients included in the study provided written consent form for their retrospective data to be collected as part of the study procedures. Statistical analysis is reported in **S1 Appendix**.

## 2.5 Statistical analysis

The distribution of variables was explored using the Shapiro-Wilk test. Continuous variables were reported as medians and interquartile range (IQR), while categorical variables were expressed as relative frequencies and percentages. Binary logistic regression was used to generate a formula for predicting significant fibrosis secondary to MASLD–BIMAST score–based on variables which were significantly different on univariate analysis. Calibration was estimated as the Brier score, while goodness-of-fit as the Hosmer-Lemeshow test. Univariate analysis was carried out using Mann-Whitney for continuous, and chi-square test for categorical variables respectively. Kruskal-Wallis or ANOVA with post-hoc corrections was used for comparison between multiple groups. Receiver operating characteristic (ROC) curves were used to assess the diagnostic performance of the derived formula compared to traditional screening methods for fibrosis assessment. Areas under ROC curve (AUROC) with 95% confidence intervals were calculated under nonparametric (distribution free) assumption. Optimal cut-off values were calculated to maximise sensitivity and specificity, while positive predictive value (PPV) and negative predictive value (NPV) were reported based upon the observed prevalence of liver disease within the population. Finally, pairwise statistical comparison of AUROCs was performed using the DeLong method between the derived score and traditional screening methods. Sample size was estimated at 400 patients. However, due to covid restrictions only 300 were enrolled.

All tests were two-sided and a P value 0.05 was considered significant. Statistical analysis was performed using SPSS© (version 24.0; SPSS Inc Chicago, IL).

## 3. Results

### 3.1 Study population

Between April 2019 and January 2021, a total of 300 patients with T2DM were enrolled from the North-West London GP network. Overall, 287 patients underwent the whole screening procedure, while 13 did not complete the screening, and were excluded. The study population was diverse in terms of ethnic background and in terms of severity of T2DM and anti-diabetic treatments. Demographic and clinical characteristics of the study population are shown in **S9 and S10 Tables in S1 Appendix**.

The overall prevalence of MASLD, based on US and CAP score, was 64% (186/287), while the prevalence of other liver diseases was 9% (28/287: 27 MetALD and 1 with chronic hepatitis B). As per TE, the prevalence of significant liver disease was 17% (50/287), the prevalence of advanced fibrosis was 10% (31/287), and the prevalence of newly diagnosed cirrhosis was 3% (8/287). Due to the COVID-19 related restrictions, only 11 patients underwent a liver biopsy among those with elevated LSM: all the biopsied cases had liver fibrosis stage $\geq$2 as per CRN scoring system. Among those with LSM$\geq$8.1 kPa, 4 patients had fibrosis stage 2, 5 patients fibrosis stage 3 and 2 patients fibrosis stage 4. Among those with LSM$\geq$12.1 kPa, 3 patients had fibrosis stage 2, 3 patients fibrosis stage 3 and 2 patients fibrosis stage 4.

### 3.2 Derivation of the BIMAST score and internal validation

Within the MASLD cohort (n = 186), those with significant fibrosis (LSM $\geq$8.1 kPa, n = 50) had higher body mass index (BMI) (36.8 vs 30.3 kg/m2, p = 0.0001), waist circumferences (120 vs 105 cm, p = 0.0001), and higher aspartate aminotransferase (AST, 37 vs 26 IU/L, p = 0.0001), compared to those with normal stiffness (n = 136) (**S10 and S11 Tables in S1 Appendix**).

Table 1. Diagnostic performance of the BIMAST in the derivation and validation cohorts.

| BIMAST | Calibration (Brier score) | Goodness of fit (Hosmer-Lemeshow test) | AUROC (95%CI, p-value) for signicant fibrosis | AUROC (95%CI, p-value) for advanced fibrosis |
|---|---|---|---|---|
| Derivation cohort | 0.12 | 0.9 | 0.81 (95%CI: 0.72–0.9, p<0.0001) | 0.84 (95%CI: 0.72–0.95, p<0.0001) |
| Internal validation cohort | 0.13 | 0.89 | 0.91 (95%CI: 0.82–0.99, p<0.0001) | 0.908 (95%CI: 0.81–0.99, p<0.0001) |
| Royal free cohort | 0.31 | 0.67 | 0.7 (95%CI: 0.63–0.77, p<0.0001) | 0.72 (95%CI: 0.65–0.8, p<0.0001) |
| Sicilian cohort | 0.38 | 0.6 | 0.608 (95%CI: 0.5–0.71, p = 0.037) | 0.602 (95%CI: 0.51–0.69, p = 0.0001) |

The whole study population was split into a derivation (n = 194) and a validation (n = 93) cohort, following a 2:1 random allocation. The derivation and the internal validation cohorts were similar in terms of clinical features (**S12 Table in S1 Appendix**). On multivariate analysis, BMI, waist circumference, AST and education rank were independently associated with the presence of significant fibrosis (**S13 Table in S1 Appendix**). However, only BMI and AST were moved forward to generate a new score, called the **BIMAST score**. Waist circumference and education rank were omitted to increase the potential usability of the score, which was computed as:

$$BIMAST\ score:\ 0.17*(BMI,\ kg/m^2)\ +\ 0.054*(AST,\ IU/L)\ -\ 8.771$$

The BIMAST score can be calculated online using the platform: https://callbuddy.eu/BIMAST/index.html.

The Hosmer-Lemeshow test and Brier score for the BIMAST score were 0.9 and 0.12, confirming that the derived model fitted well the derivation cohort and had good calibration (**Table 1**). In the derivation cohort, the BIMAST score was able to predict the presence of significant fibrosis (LSM≥8.1 kPa, n = 33) accurately, with an AUROC of 0.81 (95%CI: 0.72–0.9, p<0.0001) (**Fig 1A**). A cut-off of 0.063 gave 94% sensitivity and 44% specificity, with PPV 22%

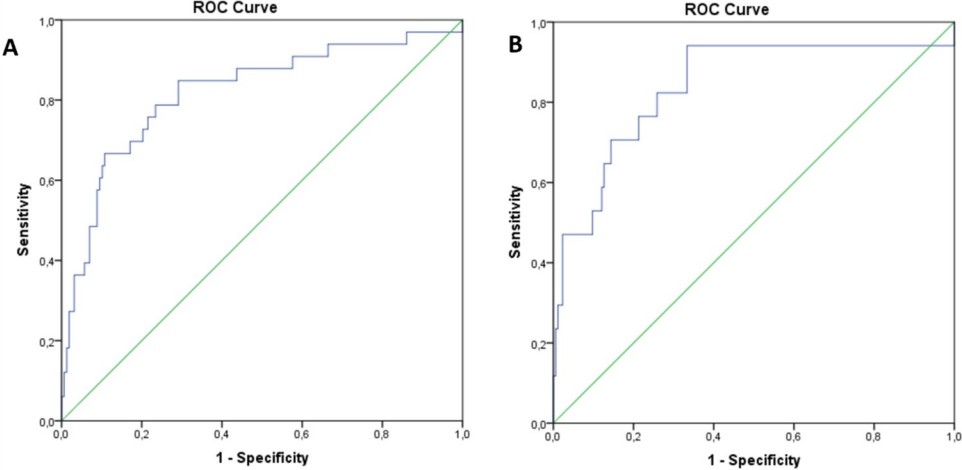

**Fig 1. Diagnostic performance of the BIMAST score for predicting significant and advanced fibrosis in the derivation cohort (diabetes primary care).** The figure illustrates the receiver operating characteristic curve of the BIMAST score for predicting LSM ≥ 8.1 kPa (Fig 1A) and for predicting LSM ≥ 12.1 kPa (Fig 1B) in the derivation cohort (diabetes primary care).

and NPV 97% for significant fibrosis. Moreover, the BIMAST score was able to predict the presence of advanced fibrosis (LSM≥12.1 kPa, n = 17) accurately, with an AUROC of 0.84 (95%CI: 0.72–0.95, p<0.0001) (**Fig 1B**). A cut-off of 0.102 carried sensitivity 94%, specificity 50%, positive predictive value (PPV) 20% and negative predictive value (NPV) 99% for advanced fibrosis.

### 3.3 Internal validation

In the internal validation cohort, the BIMAST score showed a Hosmer-Lemeshow test and Brier score of 0.89 and 0.13, confirming that the derived model fitted well the validation cohort well and had good calibration (**Table 1**). In this cohort, the BIMAST score was able to predict the presence of significant fibrosis (LSM≥8.1 kPa, n = 17) accurately, with an AUROC 0.91 (95%CI: 0.82–0.99, p<0.0001) (**Fig 1A**). A cut-off of 0.063 gave 70% sensitivity and 90% specificity, with PPV 63% and NPV 94% for significant fibrosis. Similarly, the BIMAST score predicted the presence of advanced fibrosis (LSM≥12.1 kPa, n = 14) accurately, with an AUROC 0.908 (95%CI: 0.81–0.99, p<0.0001) (**Fig 1B**). A cut-off of 0.102 carried sensitivity 63%, specificity 95%, PPV 66% and NPV 99% for advanced fibrosis.

### 3.4 BIMAST score versus other screening strategies

In the whole primary care cohort, when compared to other screening strategies, the BIMAST score performed better. The AUROC curves for diagnosing significant fibrosis were 0.74 (95% CI: 0.66–0.83, p<0.0001) for US plus LFTs, 0.72 (95%CI: 0.65–0.8, p = 0<0.0001) for NAFLD fibrosis score, 0.55 (95%CI: 0.43–0.67, p = 0.33), 0.55 (95%CI: 0.43–0.67, p = 0.33) for ELF and 0.57 (95%CI: 0.46–0.68, p = 0.21) for FIB-4. The pairwise comparison of AUROC curves (DeLong method) demonstrated that the BIMAST score was better than US plus LFTs (p = 0.01), NAFLD fibrosis score (p = 0.009), ELF test (p<0.0001) and FIB-4 (p<0.0001) in diagnosing the presence of significant fibrosis in the community (**Fig 2A**). Similarly, the BIMAST score outperformed US plus abnormal LFTs (p = 0.01), NAFLD fibrosis score

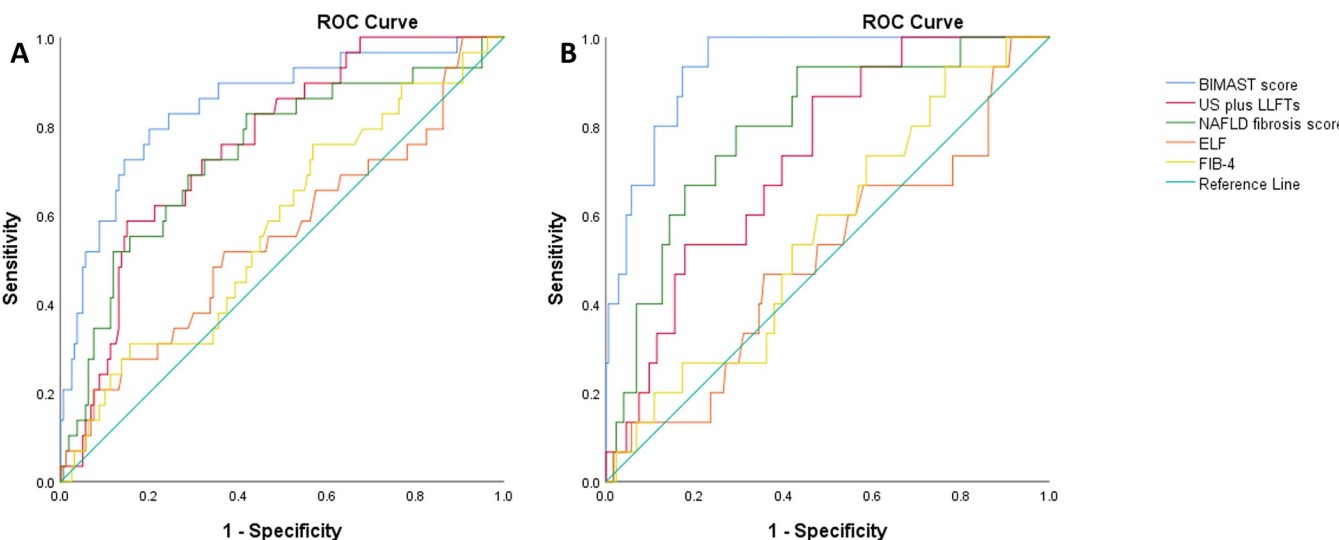

**Fig 2. BIMAST score vs conventional screening methods for predicting significant and advanced fibrosis in the diabetic primary care population (whole study population).** The figure illustrates the receiver operating characteristic curve of the BIMAST score vs conventional methods for predicting LSM ≥ 8.1 kPa (Fig 2A) and for predicting LSM ≥ 12.1 kPa (Fig 2B) in the whole study population.

(p<0.0001), ELF test (p<0.0001) and FIB-4 (p<0.0001) for diagnosing the presence of advanced fibrosis (DeLong method) (**Fig 2B**).

### 3.5 False negatives at screening

Overall, 19 (19/50 = 38%) patients with LSM≥8.1 kPa and 11 (11/31 = 35%) patients with LSM ≥12.1 kPa were missed by FIB-4 (false negatives). Similarly, 27 (27/50 = 54%) patients with LSM≥8.1 kPa and 15 (15/31 = 50%) patients with LSM≥12.1 kPa were missed by ELF (false negatives). Notably, only 5 (5/50% = 10%) patients with LSM ≥8.1 kPa and 3 (3/31 = 10%) patients with LSM≥12.1 kPa were missed by BIMAST (false negatives).

### 3.6 External validation of the BIMAST score

Both the Royal Free cohort (n = 218) and the Sicilian cohort (n = 168) presented higher LFTs and LSM values compared to the primary care cohort (**S14 Table and S1 Fig in S1 Appendix**).

In the Royal Free cohort, the Hosmer-Lemeshow test and the Brier score for the BIMAST score were 0.67 and 0.31, suggesting that the BIMAST score had a moderate goodness-of-fit and calibration (**Table 1**). Specifically, the BIMAST score predicted significant fibrosis (LSM≥ 8.1 kPa, n = 105) with an AUROC of 0.7 (95%CI: 0.63–0.77, p<0.0001), while a cut-off of the BIMAST of 0.063 gave sensitivity 34%, specificity 91%, PPV 76% and NPV 40% (**Fig 3A**). Furthermore, the BIMAST predicted advanced fibrosis (LSM≥ 12.1 kPa, n = 66) with an AUROC of 0.72 (95%CI: 0.65–0.8, p<0.0001), while a cut-off of 0.102 gave sensitivity 43%, specificity 89%, PPV 62% and NPV 23%. vs 0.68 (95%CI: 0.6–0.76, p<0.0001) of FIB-4 (**Fig 3B**). The pairwise comparison between AUROC curves (De Long method) confirmed that the BIMAST and the FIB-4 performed similarly in this cohort. Finally, 35 (35/105 = 33%) patients with LSM≥8.1 kPa and 18 (18/68 = 26%) patients with LSM≥12. 1 kPa had normal FIB-4 (false negatives).

In the Sicilian cohort, the Hosmer-Lemeshow test and the Brier score for the BIMAST score were 0.6 and 0.38, suggesting that the BIMAST score had a moderate goodness-of-fit and calibration (**Table 1**). Specifically, the BIMAST score predicted LSM≥ 8.1 kPa (n = 114) with an AUROC of 0.608 (95%CI: 0.5–0.71, p = 0.037), while a cut-off of 0.063 gave sensitivity 27%, specificity 86%, PPV 83% and NPV 30% (**Fig 4A**). Moreover, the BIMAST score predicted LSM≥ 12.1 kPa (n = 65) with an AUROC of 0.602 (95%CI: 0.51–0.69, p = 0.0001), while a cut-

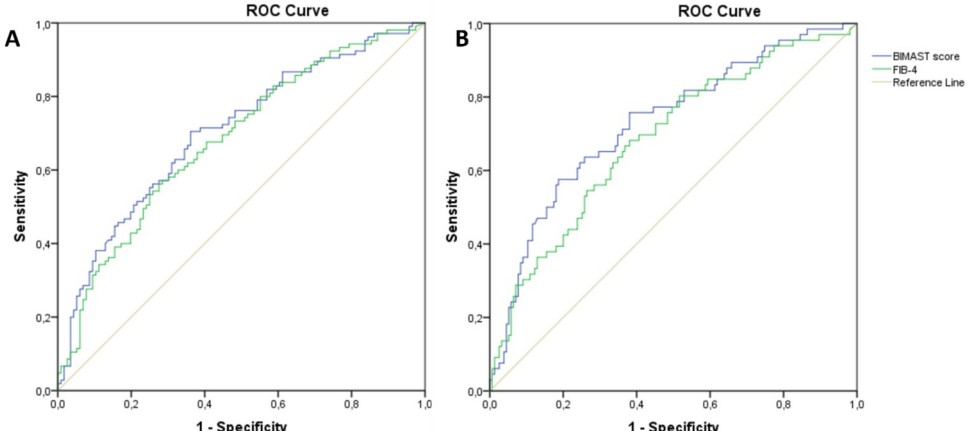

**Fig 3. BIMAST score vs FIB-4 for predicting significant and advanced fibrosis in the Royal Free cohort.** The figure shows the receiver operating characteristic curve of BIMAST score vs FIB-4 for predicting LSM≥ 8.1 kPa (Fig 3A) and LSM ≥12.1 kPa (Fig 3B) in the Royal Free cohort.

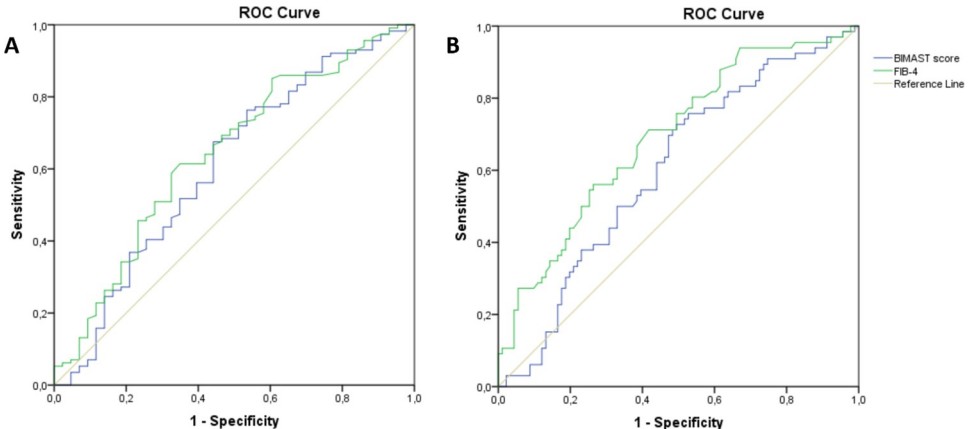

**Fig 4. BIMAST score vs FIB-4 for predicting significant and advanced fibrosis in the Sicilian cohort.** The figure shows the receiver operating characteristic curve of BIMAST score vs FIB-4 for predicting LSM≥ 8.1 kPa (Fig 4A) and LSM ≥12.1 kPa (Fig 4B) in the Sicilian cohort.

off of 0.102 gave sensitivity 20%, specificity 85%, PPV 48% and NPV 40%.vs 0.69 (95%CI: 0.609–0.77, p = 0.0001) of FIB-4 (**Fig 4B**). The pairwise comparison between AUROC curves (De Long method) revealed that FIB-4 performed better than the BIMAST score in this cohort. Finally, 47 (46/114 = 41%) patients with LSM≥8.1 kPa and 20 (20/67 = 29%) patients with LSM≥12.1 kPa had normal FIB-4 (false negatives).

### 3.7 Cost-effectiveness analysis

Overall, all screening strategies were associated with QALY gains, ranging from 121–149 years, with TE (148.73 years) resulting in the most substantial gains, followed by BIMAST score (141.01 years), FIB-4 (134.07 years), ELF (131.68 years) and NAFLD fibrosis score (121.25 years). The ICER of BIMAST score and TE compared to SOC were £2,337.92 and £2,480 per QALY gained, respectively (**Table 2**).

The ICER was most sensitive to variations in progression rates (effect of early diagnosis on disease progression), screening test sensitivity and specificity, and model time horizon. Whereas all screening strategies were found to be cost-effective compared to standard of care in the base-case, when the time horizon was decreased from 40 years (lifetime) to 5 years, only BIMAST and FIB-4 remained cost-effective within the NICE cost-effectiveness threshold criteria (**Fig 5**).

## 4. Discussion

Metabolic-dysfunction associated steatotic liver disease (MASLD) is now the major cause of liver disease in Western countries and is expected to become the first indication for liver transplantation in the United States in the next decade [20]. The Lancet commission for tackling liver disease indicated that the definition and implementation of models of care for early detection, represents an area of priority [21]. Moreover, several studies have highlighted a significant gap between guidelines and *real-life* clinical approach, not only across different continents [22] but also within Europe [23]. Specifically, primary care physicians demand a clearly defined, pragmatical referral management pathway, which can easily be implemented in high-risk groups in the community. Among others, patients with T2DM are at higher risk for advanced liver disease and are, therefore, an ideal target for MASLD screening [5, 6].

**Table 2. Base-case cost-effectiveness analysis of MASLD screening strategies versus standard of care (baseline screening).**

| *Screening strategy 1: US + LFTs* | Screening | Standard of care |
|---|---|---|
| Discounted life expectancy, entire cohort (years) | 3,751.88 | 3,594.85 |
| QALYs gained, entire cohort (years) | 138.19 | - |
| Increase in correct diagnoses compared to baseline screening (%) | 10.73 | - |
| Lifetime discounted per person cost (£) | 13,542.93 | 12,295.53 |
| Incremental cost per person (£) | 0.0008 | - |
| **Incremental cost-effectiveness ratio (£/QALY)** | **2,337.92** | - |
| *Screening strategy 2: FIB-4* | | |
| Discounted life expectancy, entire cohort (years) | 3,747.62 | 3,594.85 |
| QALYs gained, entire cohort (years) | 134.07 | - |
| Increase in correct diagnoses compared to baseline screening (%) | 8,29 | - |
| Lifetime discounted per person cost (£) | 13,361.87 | 12,295.53 |
| Incremental cost per person (£) | 0.0009 | - |
| **Incremental cost-effectiveness ratio (£/QALY)** | **2,059.98** | - |
| *Screening strategy 3: NAFLD fibrosis score* | | |
| Discounted life expectancy, entire cohort (years) | 3,734.50 | 3,594.85 |
| QALYs gained, entire cohort (years) | 121.25 | - |
| Increase in correct diagnoses compared to baseline screening (%) | -2.32 | - |
| Lifetime discounted per person cost (£) | 13,275.08 | 12,295.53 |
| Incremental cost per person (£) | 0.0010 | - |
| **Incremental cost-effectiveness ratio (£/QALY)** | **2,092.47** | - |
| *Screening strategy 4: BIMAST score* | | |
| Discounted life expectancy, entire cohort (years) | 3,754.88 | 3,594.85 |
| QALYs gained, entire cohort (years) | 141.01 | - |
| Increase in correct diagnoses compared to baseline screening (%) | 10.76 | - |
| Lifetime discounted per person cost (£) | 13,366.51 | 12,295.53 |
| Incremental cost per person (£) | 0.0009 | - |
| **Incremental cost-effectiveness ratio (£/QALY** | **1,967.11** | - |
| *Screening strategy 5: ELF test* | | |
| Discounted life expectancy, entire cohort (years) | 3,745.06 | 3,594.85 |
| QALYs gained, entire cohort (years) | 131.68 | - |
| Increase in correct diagnoses compared to baseline screening (%) | 8.48 | - |
| Lifetime discounted per person cost (£) | 13,587.54 | 12,295.53 |
| Incremental cost per person (£) | 0.0008 | - |
| **Incremental cost-effectiveness ratio (£/QALY)** | **2,541.24** | - |
| *Screening strategy 6: Transient elastography* | | |
| Discounted life expectancy, entire cohort (years) | 3,762.89 | 3,594.85 |
| QALYs gained, entire cohort (years) | 148.73 | - |
| Increase in correct diagnoses compared to baseline screening (%) | 15.05 | - |
| Lifetime discounted per person cost (£) | 13,717.67 | 12,295.53 |
| Incremental cost per person (£) | 0.0007 | - |
| **Incremental cost-effectiveness ratio (£/QALY)** | **2,476.57** | - |

Abbreviations: US: ultrasound; LFTs: liver function tests; ELF test: Enhanced liver fibrosis test.

In this study, we present a cohort of diabetic patients who were screened for MASLD and fibrosis in primary care, without any *a priori* selection. This cohort reflects the population of a major urban city and is very diverse in terms of ethnic and socioeconomic background. This

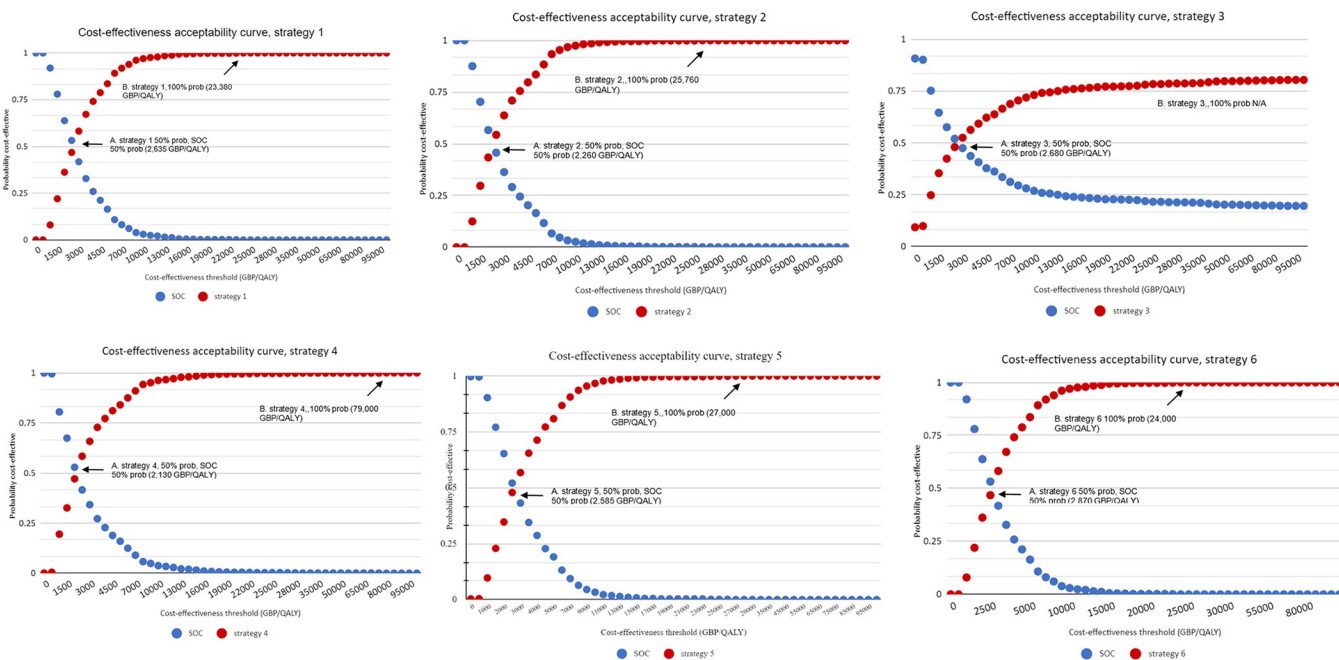

**Fig 5. Cost-effectiveness acceptability curve (strategy 1 to 6).** Red and blue lines represent the cost-effectiveness acceptability of the screening strategy and SOC, respectively. Each dot on the graph shows the probability of strategy 4 being cost-effective (Y-axis) at a given cost-effectiveness threshold (X-axis). 1. point A shows the point at which both strategies have 50% probability of being cost-effective; 2. point B shows the point at which scenario 1 has 100% probability of being cost-effective. SOC: standard of care.

study population also includes patients with a wide range of antidiabetic treatments, glycaemic control and length of disease. Overall, the prevalence of MASLD based on US was 64%, while the prevalence of significant liver disease and cirrhosis secondary to MASLD were 17% and 3% respectively. Among other factors, visceral obesity, education level and AST were the main predictors for the presence of significant liver disease in primary care.

According to the latest published EASL guidelines, patients with T2DM should be screened for MASLD using a two-tier system, i.e. FIB-4 and/or NAFLD fibrosis score in primary care, followed by ELF and/or TE in a specialist setting [10]. However, the performance of blood-based non-invasive markers remains largely unexplored, with the latest AALD guidelines hinting at possible underestimation of fibrosis among diabetics [24]. Recent evidence has raised concerns that FIB-4 accuracy is much lower in the community, with high false negative rates [13, 25, 26]. We confirm here that 38% of patients with LSM$\geq$8.1 kPa and 35% of patients with LSM $\geq$12.1 kPa were missed by FIB-4.

From an epidemiological perspective, the *spectrum effect* describes the variation in the diagnostic performance of predictive tests when applied to populations with different disease prevalence. Ideally, new screening tests should be derived from a population which mirrors the target population for the test, so that *spectrum biases* could be minimised [27]. We developed the BIMAST score based on the clinical predictors of significant fibrosis in this cohort (BMI and AST). The BIMAST score predicts the presence of significant and advanced fibrosis accurately and is easy to calculate in a primary care setting. We also demonstrated that using BIMAST is cost-effective and that performs better than other screening strategies. Moreover, the BIMAST is characterised by an elevated NPV, therefore acting as an excellent screening test, and reduces the rate of false negatives from 38% to 10% compared to FIB-4. When validated internally, the BIMAST score showed an excellent diagnostic performance. Nevertheless,

this score did not appear to perform as well in the two external cohorts, but this is likely due to *spectrum biases*. The phenotype of the patients included in the Royal Free and the Sicilian cohort was typical of those observed in secondary/tertiary care and therefore with higher prevalence of severe liver disease compared to the community. Notably, the BIMAST score showed a good calibration and a moderate goodness-of-fit in both the validation cohorts. Collectively, our results suggest that the BIMAST score and the FIB-4 are both victims of spectrum effect but in an opposite way.

This study has some limitations. Firstly, we used TE as the gold standard as only 11 patients were biopsied. However, several other studies support the use of elastographic techniques, when screening the general population [28]. Furthermore, even only 11, these biopsies' staging was in absolute agreement with the stiffness results. Nevertheless, applying TE as reference for the economic model might translated into an overestimation of the cost-effectiveness of the TE in this scenario. Secondly, the external validation cohorts were not from primary care but from secondary/tertiary care. Future work should focus on validating the BIMAST score on external community cohorts and on evaluating the effect of earlier diagnosis on extra-hepatic complications, such as cardiovascular disease.

To summarise, in this study, we developed and validated the BIMAST score, an easy to calculate score which includes BMI and AST. The BIMAST score is cost-effective and predicts the presence of significant and advanced fibrosis accurately in the community. Future studies should focus on testing the BIMAST score in primary care in people with diabetes.

## Supporting information

**S1 Appendix.**
(DOCX)

## Author Contributions

**Conceptualization:** Mark R. Thursz, Pinelopi Manousou.

**Data curation:** Roberta Forlano, Tijana Stanic, Sahan Jayawardana, Salvatore Petta, Emmanouil Tsochatzis.

**Formal analysis:** Roberta Forlano, Tijana Stanic, Sahan Jayawardana.

**Funding acquisition:** Pinelopi Manousou.

**Investigation:** Roberta Forlano.

**Methodology:** Roberta Forlano.

**Supervision:** Mark R. Thursz, Pinelopi Manousou.

**Validation:** Salvatore Petta, Emmanouil Tsochatzis.

**Writing – original draft:** Roberta Forlano, Tijana Stanic, Sahan Jayawardana.

**Writing – review & editing:** Benjamin H. Mullish, Michael Yee, Elias Mossialos, Robert D. Goldin, Salvatore Petta, Emmanouil Tsochatzis, Mark R. Thursz, Pinelopi Manousou.

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
