## [Decision Letter · Decision Letter 0]

4 Jun 2024

PONE-D-24-15954Derivation and validation of the BIMAST score for predicting the presence of fibrosis due to Metabolic dysfunction-associated steatotic liver disease among diabetic patients in the community.PLOS ONE

Dear Dr. Manousou,

Thank you for submitting your manuscript to PLOS ONE. After careful consideration, we feel that it has merit but does not fully meet PLOS ONE’s publication criteria as it currently stands. Therefore, we invite you to submit a revised version of the manuscript that addresses the points raised during the review process.

We look forward to receiving your revised manuscript.

Kind regards,

Ashraf Elbahrawy

Academic Editor

PLOS ONE

Journal Requirements:

"RF is the recipient of the EASL PhD fellowship Juan Rodes 2018. BHM is the recipient of an NIHR Academic Clinical Lectureship (CL-2019-21-002). The Division of Digestive Diseases receives financial support from the National Institute of Health Research (NIHR) Imperial Biomedical Research Centre (BRC). "

3. Please expand the acronym “EASL and NIHR” (as indicated in your financial disclosure) so that it states the name of your funders in full.

"no"

Reviewers' comments:

Reviewer's Responses to Questions

**Comments to the Author**

1. Is the manuscript technically sound, and do the data support the conclusions?

Reviewer #1: Yes

Reviewer #2: Yes

2. Has the statistical analysis been performed appropriately and rigorously? 

Reviewer #1: Yes

Reviewer #2: Yes

3. Have the authors made all data underlying the findings in their manuscript fully available?

Reviewer #1: Yes

Reviewer #2: Yes

4. Is the manuscript presented in an intelligible fashion and written in standard English?

Reviewer #1: Yes

Reviewer #2: No

5. Review Comments to the Author

Reviewer #1: I reviewed your manuscript titled “Derivation and validation of the BIMAST score for predicting the presence of fibrosis due to Metabolic dysfunction-associated steatotic liver disease among diabetic patients in the community.”. This study is well written, nicely designed and highly boosted with illustrative figures. Actually, this study provides a novel, simple and cheap method for easy prediction of the MASLD and its prognosis in diabetics, particularly at the primary care level, which will greatly affect clinical practice. However, some clarifications and revisions would help to better understand and put in perspective your findings:

- By reading the previous work of the authors, Reference 13 (attached file); I think the same dataset of the previous publication was used for the current derivation of the developed BIMAST score, and even some figures are the same. However, no clear statement that the same dataset was used in both publications, and I think the same figures may need a copyright agreement for re-publication. Moreover, both papers have the same research ethics committee approval no. (REC approval 18/LO/1742, IRAS 251274)).

So, this needs clarification and clear statements in the manuscript.

- Methods of the diagnosis of MASLD, which you have already mentioned that it is challenging,

were defined clearly in the supplementary files, however, missed from the main manuscript. I think it is crucial to be mentioned as part of the study design and methodology.

- In the methodology you mentioned that Standard of care (abnormal LFTs or no screening), rather than the gold standard liver histology, was used as a standard test for different methods including BIMAST score in prediction of fibrosis stage. Nonetheless, diagnosing MASLD relying only on abnormal liver enzymes carry the risk for false negative cases which may carry advanced fibrosis. Moreover, In both external validation cohorts, you mentioned that, MASLD was diagnosed based on either US or histology. So, clarification may be needed on this point.

Another point, in the discussion you mentioned that one of the limitations was using TE rather than liver biopsy as a gold standard! while TE was one of the different methods used for comparison with the standard test!

this contradiction needs clarification.

- Lastly, all figures are hazy and overlying data are not clear (especially figure 5).

Reviewer #2: The study by Forlano et al. is an intriguing validation study of a new score to non-invasively diagnose MASLD and fibrosis in patients with diabetes.

The aim and the findings of this study represent a hot topic in the field of hepatology, and it is well documented in both the introduction and discussion.

From a statistical point of view, the investigation is rigorously performed. however, I recommend a deep English revision and to address some issues.

- Line 94: please explain what does ELF stands for

- In material and methods section: please explain how does MASLD was diagnosed. According to 2024 consensus statement MASLD diagnosis required the presence of cardiometabolic risk factors in addiction to liver steatosis.

- Line 226: what does “significant liver disease” mean?

- What about antidiabetic therapy? Since GLP1-RA have been shown to improve liver steatosis, were there any patients undergoing antidiabetic therapy with these drugs?

- In the abstract: please clarify to which cohort the percentages presented in the results section are referred to.

6. PLOS authors have the option to publish the peer review history of their article (what does this mean?). If published, this will include your full peer review and any attached files.

Reviewer #1: **Yes: **Hassan Atalla, MD

Reviewer #2: No

---

## [Author Response · Author response to Decision Letter 0]

12 Jun 2024

Re: PONE-D-24-15954. Derivation and validation of the BIMAST score for predicting the presence of fibrosis due to Metabolic dysfunction-associated steatotic liver disease among diabetic patients in the community.

Dear Editorial Team,

Thank you for giving us the opportunity to submit a revised draft of the manuscript “Derivation and validation of the BIMAST score for predicting the presence of fibrosis due to Metabolic dysfunction-associated steatotic liver disease among diabetic patients in the community” for publication in PLOS one. We appreciate the time and effort that yourself and the peer reviewers dedicated to providing feedback on our manuscript and are grateful for the insightful comments on and valuable improvements to our paper. 

We have made considerable changes in light of the reviewer comments, which we feel address their queries, and which strengthen the manuscript; we hope as such that it is now acceptable for publication. Please see below for a point-by-point response to the reviewers’ comments and concerns.

Reviewer n. 1: 

I reviewed your manuscript titled “Derivation and validation of the BIMAST score for predicting the presence of fibrosis due to Metabolic dysfunction-associated steatotic liver disease among diabetic patients in the community.”. This study is well written, nicely designed and highly boosted with illustrative figures. Actually, this study provides a novel, simple and cheap method for easy prediction of the MASLD and its prognosis in diabetics, particularly at the primary care level, which will greatly affect clinical practice. However, some clarifications and revisions would help to better understand and put in perspective your findings:

- By reading the previous work of the authors, Reference 13 (attached file); I think the same dataset of the previous publication was used for the current derivation of the developed BIMAST score, and even some figures are the same. However, no clear statement that the same dataset was used in both publications, and I think the same figures may need a copyright agreement for re-publication. Moreover, both papers have the same research ethics committee approval no. (REC approval 18/LO/1742, IRAS 251274)).

So, this needs clarification and clear statements in the manuscript.

Response: We thank the reviewer for their comment. We have made clearer the use of the same dataset as previous publication within the text (page 7, line 118):

“Of note, this study population was previously published”.

With regards to figures, we thank the reviewer for raising the issue. As previous publication was open access and granted Creative Commons CC BY license, there is no need for permission to re-use the article contents.

- Methods of the diagnosis of MASLD, which you have already mentioned that it is challenging,

were defined clearly in the supplementary files, however, missed from the main manuscript. I think it is crucial to be mentioned as part of the study design and methodology.

Response: We agree with the reviewer for their comment. We have included how MASLD was diagnosed in the main text (page 7, line 119 to 121). “MASLD was defined as presence of steatosis with at least one cardiometabolic risk factor and in absence of other causes of liver disease.” 

Of note, all patients have a cardiometabolic risk factors as they suffer from Type-2 Diabetes Mellitus.

- In the methodology you mentioned that Standard of care (abnormal LFTs or no screening), rather than the gold standard liver histology, was used as a standard test for different methods including BIMAST score in prediction of fibrosis stage. Nonetheless, diagnosing MASLD relying only on abnormal liver enzymes carry the risk for false negative cases which may carry advanced fibrosis. Moreover, In both external validation cohorts, you mentioned that, MASLD was diagnosed based on either US or histology. So, clarification may be needed on this point.

Response: We agree with the reviewer that MASLD diagnosis cannot rely on abnormal liver function tests as these are not sensitive. Nevertheless, MASLD screening among general practitioners still relies on abnormal liver function tests and/or no screening at all. We have clarified this in the main manuscript (page 8, line 127-129):

“Definition of standard of care for MASLD screening in primary care was obtained from previously published economic evaluations on MASLD screening (Tanajewski et al., 2017; Thavorn & Coyle, 2015), and defined based on what is routinely performed in the community (either abnormal LFTs or no screening)”

Finally, external cohorts used in this study, include patients already screened and seen by the specialist, whereby MASLD was diagnosed as per guidelines (i.e. based on either US or histology).

- Another point, in the discussion you mentioned that one of the limitations was using TE rather than liver biopsy as a gold standard! while TE was one of the different methods used for comparison with the standard test!

this contradiction needs clarification.

Response: We agree with the reviewer for their comment. We have emphasized this concept as limitation of the study (page 21, line 417 to 419):

“Nevertheless, applying TE as reference for the economic model might be translated into an overestimation of the cost-effectiveness of the TE in this scenario.”

- Lastly, all figures are hazy and overlying data are not clear (especially figure 5).

Response: We agree with the reviewer for their comment. We have submitted a better version of the figures.

Reviewer n. 2: 

The study by Forlano et al. is an intriguing validation study of a new score to non-invasively diagnose MASLD and fibrosis in patients with diabetes.

The aim and the findings of this study represent a hot topic in the field of hepatology, and it is well documented in both the introduction and discussion.

From a statistical point of view, the investigation is rigorously performed. However, I recommend a deep English revision and to address some issues.

- Line 94: please explain what does ELF stands for

Response: We agree with the reviewer for their comment. We have clarified the acronym in “Enhanced liver fibrosis”

- In material and methods section: please explain how does MASLD was diagnosed. According to 2024 consensus statement MASLD diagnosis required the presence of cardiometabolic risk factors in addiction to liver steatosis.

Response: We agree with the reviewer for their comment. We have modified the methods as per reviewer n.1 comment.

- Line 226: what does “significant liver disease” mean?

 Response: We have included a definition in the methods (page 7, line 119):

“Significant liver disease was defined as LSM≥8.1 kPa.”

- What about antidiabetic therapy? Since GLP1-RA have been shown to improve liver steatosis, were there any patients undergoing antidiabetic therapy with these drugs?

Response: We agree with the reviewer that antidiabetic therapy is an important point to consider when analysing risk for MASLD in diabetics. 

In this cohort, 37 (13%) patients were on a GLP-1R agonist with a median length of disease of 11 (4-16) years. There was no difference in terms of anti-diabetic treatment as well as length of the disease between those with and without significant liver disease. Results were also shown in our previously published paper (Forlano et al., 2023).

- In the abstract: please clarify to which cohort the percentages presented in the results section are referred to.

Response: We have clarified the population used for descriptive statistic (page 3, line 59):

“In the whole population, …”

REFERENCES

Forlano, R., Stanic, T., Jayawardana, S., Mullish, B. H., Yee, M., Mossialos, E., Goldin, R., Petta, S., Tsochatzis, E., Thursz, M., & Manousou, P. (2023). A prospective study on the prevalence of MASLD in people with type-2 diabetes in the community. Cost effectiveness of screening strategies. Liver Int. https://doi.org/10.1111/liv.15730

Tanajewski, L., Harris, R., Harman, D. J., Aithal, G. P., Card, T. R., Gkountouras, G., Berdunov, V., Guha, I. N., & Elliott, R. A. (2017). Economic evaluation of a community-based diagnostic pathway to stratify adults for non-alcoholic fatty liver disease: a Markov model informed by a feasibility study. BMJ Open, 7(6), e015659. https://doi.org/10.1136/bmjopen-2016-015659

Thavorn, K., & Coyle, D. (2015). Transient Elastography and Controlled Attenuation Parameter for Diagnosing Liver Fibrosis and Steatosis in Ontario: An Economic Analysis. Ont Health Technol Assess Ser, 15(19), 1-58. https://www.ncbi.nlm.nih.gov/pubmed/26664666

---

## [Decision Letter · Decision Letter 1]

8 Jul 2024

Derivation and validation of the BIMAST score for predicting the presence of fibrosis due to Metabolic dysfunction-associated steatotic liver disease among diabetic patients in the community.

PONE-D-24-15954R1

Dear Dr. Manousou,

We’re pleased to inform you that your manuscript has been judged scientifically suitable for publication and will be formally accepted for publication once it meets all outstanding technical requirements.

Kind regards,

Ashraf Elbahrawy

Academic Editor

PLOS ONE

Additional Editor Comments (optional):

Reviewers' comments:

Reviewer's Responses to Questions

**Comments to the Author**

1. If the authors have adequately addressed your comments raised in a previous round of review and you feel that this manuscript is now acceptable for publication, you may indicate that here to bypass the “Comments to the Author” section, enter your conflict of interest statement in the “Confidential to Editor” section, and submit your "Accept" recommendation.

Reviewer #1: All comments have been addressed

Reviewer #3: All comments have been addressed

2. Is the manuscript technically sound, and do the data support the conclusions?

Reviewer #1: Yes

Reviewer #3: Yes

3. Has the statistical analysis been performed appropriately and rigorously? 

Reviewer #1: Yes

Reviewer #3: Yes

4. Have the authors made all data underlying the findings in their manuscript fully available?

Reviewer #1: Yes

Reviewer #3: Yes

5. Is the manuscript presented in an intelligible fashion and written in standard English?

Reviewer #1: Yes

Reviewer #3: Yes

6. Review Comments to the Author

Reviewer #1: Thanks for addressing all the raised revision points and I think that all required questions have been answered. The topic of this paper is interesting and would add an easy method for prediction of MASLD particularly within the hands of GPs.

Reviewer #3: (No Response)

7. PLOS authors have the option to publish the peer review history of their article (what does this mean?). If published, this will include your full peer review and any attached files.

Reviewer #1: **Yes: **Hassan Atalla, MD

Reviewer #3: **Yes: **Mohammed El-Fayoumie

---

## [Editor Report · Acceptance letter]

22 Jul 2024

PONE-D-24-15954R1 

PLOS ONE

Dear Dr. Manousou, 

I'm pleased to inform you that your manuscript has been deemed suitable for publication in PLOS ONE. Congratulations! Your manuscript is now being handed over to our production team.

Kind regards, 

on behalf of

Prof. Ashraf Elbahrawy 

Academic Editor

PLOS ONE